# Reversal of the adipostat control of torpor during migration in hummingbirds

Erich R Eberts[1,2]*, Christopher G Guglielmo[3], Kenneth C Welch Jr[1,2]

[1]Department of Biological Sciences, University of Toronto Scarborough, Toronto, Canada; [2]Department of Ecological and Evolutionary Biology, University of Toronto, Toronto, Canada; [3]Department of Biology, Advanced Facility for Avian Research, University of Western Ontario, London, Canada

**Abstract** Many small endotherms use torpor to reduce metabolic rate and manage daily energy balance. However, the physiological 'rules' that govern torpor use are unclear. We tracked torpor use and body composition in ruby-throated hummingbirds (*Archilochus colubris*), a long-distance migrant, throughout the summer using respirometry and quantitative magnetic resonance. During the mid-summer, birds entered torpor at consistently low fat stores (~5% of body mass), and torpor duration was negatively related to evening fat load. Remarkably, this energy emergency strategy was abandoned in the late summer when birds accumulated fat for migration. During the migration period, birds were more likely to enter torpor on nights when they had higher fat stores, and fat gain was positively correlated with the amount of torpor used. These findings demonstrate the versatility of torpor throughout the annual cycle and suggest a fundamental change in physiological feedback between adiposity and torpor during migration. Moreover, this study highlights the underappreciated importance of facultative heterothermy in migratory ecology.

## Editor's evaluation

The authors tested the hypothesis that individual hummingbirds employ torpor as an energy-saving mechanism to facilitate migratory fattening, even when starvation is not imminent. This is a difficult hypothesis to test because of the difficulties associated with repeatedly and accurately measuring body composition in individual migratory birds. Using captive experiments, the authors provide some indication that hummingbirds use torpor during energy emergencies in the summer – rare empirical evidence supporting the hypothesis that hummingbirds use torpor to facilitate migratory fitting.

*For correspondence:
erich.eberts@mail.utoronto.ca

**Competing interest:** The authors declare that no competing interests exist.

## Introduction

Facultative hypothermia is an energy conservation strategy that allows many mammalian and some avian species to survive periods of resource unavailability or to optimize their energy budgets in certain environments or life stages (*McKechnie and Lovegrove, 2002*; *Ruf and Geiser, 2015*). During facultative hypothermia, metabolic rates and body temperatures are reduced to varying extents across species and environmental conditions (*Ruf and Geiser, 2015*). As some of the smallest avian species, hummingbirds (Trochilidae) are known for their ability to use daily torpor, a deep, short-term form of facultative hypothermia, to cope with energetic challenges they face daily and throughout their annual cycle (*Carpenter, 1974*; *Hainsworth et al., 1977*).

Studies that have investigated the use of torpor in hummingbirds in relation to food availability and body mass suggest that torpor initiation is controlled by an endogenous mechanism sensitive to an energy-store threshold (*Hainsworth et al., 1977*; *Hiebert, 1992*; *Powers et al., 2003*). This model

**eLife digest** Torpor is an energy-saving strategy used by warm-blooded animals, including birds and small mammals. Similar to hibernation, although shorter in duration, torpor is a state of minimal activity, low body temperatures and reduced metabolism that helps animals conserve energy in unfavorable conditions. Some animals use torpor to survive times when food is not readily available. Hummingbirds, for example, eat nectar all day long to meet their high energy needs, but must build fat reserves to see them through their overnight fast. If they go to sleep with too little fat, they can descend into torpor to stretch out that limited energy supply and survive until morning.

Many hummingbirds migrate to areas with warmer weather, where food remains available, for the winter months. The ruby-throated hummingbird (*Archilochus colubris*), for example, travels over 5,000 kilometers in its fall migration. Like most long-distance migrants, ruby-throated hummingbirds increase their fat stores before departing, using these stores to fuel their journey. It is thought that this bird may use torpor as a way to accelerate fat build up before its annual migration. However, it remained unclear whether hummingbirds switched from using torpor strictly in energy emergencies, to using it as strategy to prepare for migration.

To shed light on this question, Eberts, Guglielmo and Welch investigated when, why and how hummingbirds save energy using torpor during the summer, and whether there are seasonal shifts in their use of torpor coinciding with migration. Eberts, Guglielmo and Welch hypothesized that a bird would initiate daily torpor if its energy stores fall below a critical level during the night, but that they may abandon this threshold (triggering torpor at higher fat levels) in late summer as a way to spare energy and gain fat before their annual migration.

To test their hypotheses, Eberts, Guglielmo and Welch tracked body composition, food intake, energy expenditure and torpor use throughout summer in a group of captive ruby-throated hummingbirds. In the middle of the summer, the birds entered torpor and remained torpid for longer when they went to sleep with low fat stores. In late summer, however, the same birds were more likely to enter torpor at consistent times and when they had higher fat stores. Eberts, Guglielmo and Welch also observed that the more time birds spent in torpor, the more fat they gained. This suggests that in late summer, hummingbirds switch from using torpor as a survival strategy to using it to maximize energy savings before migration.

These results clearly define the physiological rules governing torpor use in hummingbirds. They also support the long-standing assumption that torpor helps migratory species save energy and accumulate fat stores before long-haul flights.

predicts that a bird will initiate torpor if its energy stores reach critically low levels with enough time remaining in the night to achieve net energy savings. Hummingbirds typically rewarm 1–2 hr before sunrise, so they rarely enter torpor after approximately 75% of the night has elapsed; even if critically low energy levels are reached late in the night, birds may avoid entering torpor at this point if the energetic benefits are outweighed by the potential costs (e.g. predation, moult delay) (*Bouma et al., 2010*; *Carr and Lima, 2013*; *Hainsworth et al., 1977*; *Hiebert, 1992*; *Hiebert, 1990*). Previous studies also suggest that the function of torpor shifts seasonally, from an energy emergency survival mechanism to an energy-storage maximization strategy during migration (*Carpenter and Hixon, 1988*; *Hiebert, 1993*). However, this threshold has not been repeatedly and accurately measured in individual birds spanning life history stages, and the relationship between torpor use and the components of body composition (fat and lean mass) remains unclear.

We explored torpor use in ruby-throated hummingbirds (*Archilochus colubris*), which breed in eastern North America in the early and mid-summer, and migrate to wintering grounds in Mexico and Central America in the late summer. In the breeding period, birds maintain relatively lean body compositions to optimize aerial agility, important for successful courtship displays and competitive interactions that allow them to maintain secure access to food resources (*Altshuler et al., 2010*; *Hou and Welch, 2016*). But like most long-distance migrants, ruby-throated hummingbirds substantially increase their body mass prior to migratory departure to fuel their journey (*Hou and Welch, 2016*).

We repeatedly and non-invasively quantified the relationship between torpor and endogenous energy stores (fat) in ruby-throated hummingbirds to investigate the underlying rules of torpor use

during the breeding and migration periods. We predicted that in the breeding period, when birds reached a low energy-store threshold before 75% of the night had elapsed, they would enter torpor to avoid complete energy depletion, and that birds that remained normothermic until this point would not enter torpor if it would not achieve net energy savings. We also predicted that this threshold would be abandoned in the late summer to facilitate premigratory fattening. Furthermore, we predicted that in the breeding period, torpor use would be primarily driven by evening fat content and the rate at which those energy reserves were used, and that in the migration period, the amount of torpor used would be driven primarily by night length, allowing the birds to spare a maximum amount of fat stores, irrespective of longer late-summer nights.

We measured the torpor use and body composition of captive adult (and one juvenile) male ruby-throated hummingbirds (n = 16; capture mass: 2.5–3.2 g) that experienced semi-natural photoperiods, on 158 focal bird-nights throughout the summer. On all days and nights, the birds experienced air temperatures of approximately 20°C (19.7°C ± 0.0°C) to control for the potential effect of air temperature as a proximate cue for torpor use. We used respirometry to calculate rates of energy expenditure and the rate of oxidation of stored fat, and quantitative magnetic resonance (QMR) to measure body composition (*Guglielmo et al., 2011*; *Lighton, 2008*). We identified the start of torpor entry and arousal by evaluating the slope of each smoothed $V_{O_2}$ trace, and we calculated the fat content at the time of torpor entry as the amount of evening fat (g) minus the estimated amount of fat expenditure prior to torpor entry, divided by the morning body mass.

We aimed to determine the rules governing the use of torpor and whether these differed throughout the summer. We evaluated changes in the relationships between evening body composition and torpor occurrence, torpor duration, time of torpor entry, fat content at torpor entry, and amount of energy expended before torpor entry, within and among periods of consistent change in body mass. We also evaluated the influence of night length on these variables. To specifically investigate the role of torpor in driving premigratory increases in body mass, we evaluated the effects of mean torpor duration and mean daily food consumption on the amount of mass gained and duration of the fattening period.

## Results

### Body mass

Throughout the study period, 13 of 16 birds exhibited relatively low morning body masses, indicative of breeding condition, until late August or early September when they substantially increased their body mass; the birds subsequently maintained high body masses. We analyzed daily changes in morning body mass and defined 'breeding', 'fattening', and 'migration' periods, respectively (Materials and methods, *Supplementary file 3*). Additionally, three 'non-fattener' birds maintained relatively constant body masses throughout the summer (*Figure 1*; *Supplementary file 1A*). QMR scans indicated that changes in body mass were driven primarily by increases in fat (r(109)=0.94, 95% CI [0.92, 0.96] p < 0.001; *Appendix 1—figure 1A*), and that body mass and lean mass were slightly negatively correlated (r(107)=−0.26, 95% CI [−0.43,–0.08], p = 0.006; *Appendix 1—figure 1B*). Individual variation in daily food consumption and presumably activity were the primary factors leading to nightly variation in evening body composition, as environmental factors such as air temperature, humidity, and food availability were consistent throughout the study period.

### Breeding period

During the mid-summer breeding period, birds maintained consistently low morning body masses (2.77 ± 0.05 g; slope: 0.00 ± 0.00 g·day$^{-1}$; p = 0.338; *Figure 1*; *Table 1*; *Supplementary file 1A*). On average, birds used torpor on 61.6% ± 11.1% of focal bird-nights (*Supplementary file 1D*). When birds started the night with less fat, they were more likely to enter torpor (slope: –0.70 ± 0.22; p = 0.001; *Appendix 1—figure 2*), entered torpor earlier in the night (slope: 0.55 ± 0.09 h$_{entry}$ ·%fat$^{-1}$; p < 0.001), and remained torpid longer (slope: –0.63 ± 0.10 hr·%fat$^{-1}$, p < 0.001; *Figure 2A*). Neither torpor propensity (slope: –0.31 ± 0.47, p = 0.452), the time of torpor entry relative to the start of the night (slope: 0.52 ± 0.30 h$_{entry}$·h$_{night}$, p = 0.054) nor torpor duration (slope: 0.52 ± 0.32 h$_{torpor}$·h$_{night}$, p = 0.077) were significantly related to night length (*Supplementary file 1D*). Furthermore, when birds started the night with greater fat content, they expended more energy before initiating torpor (slope:

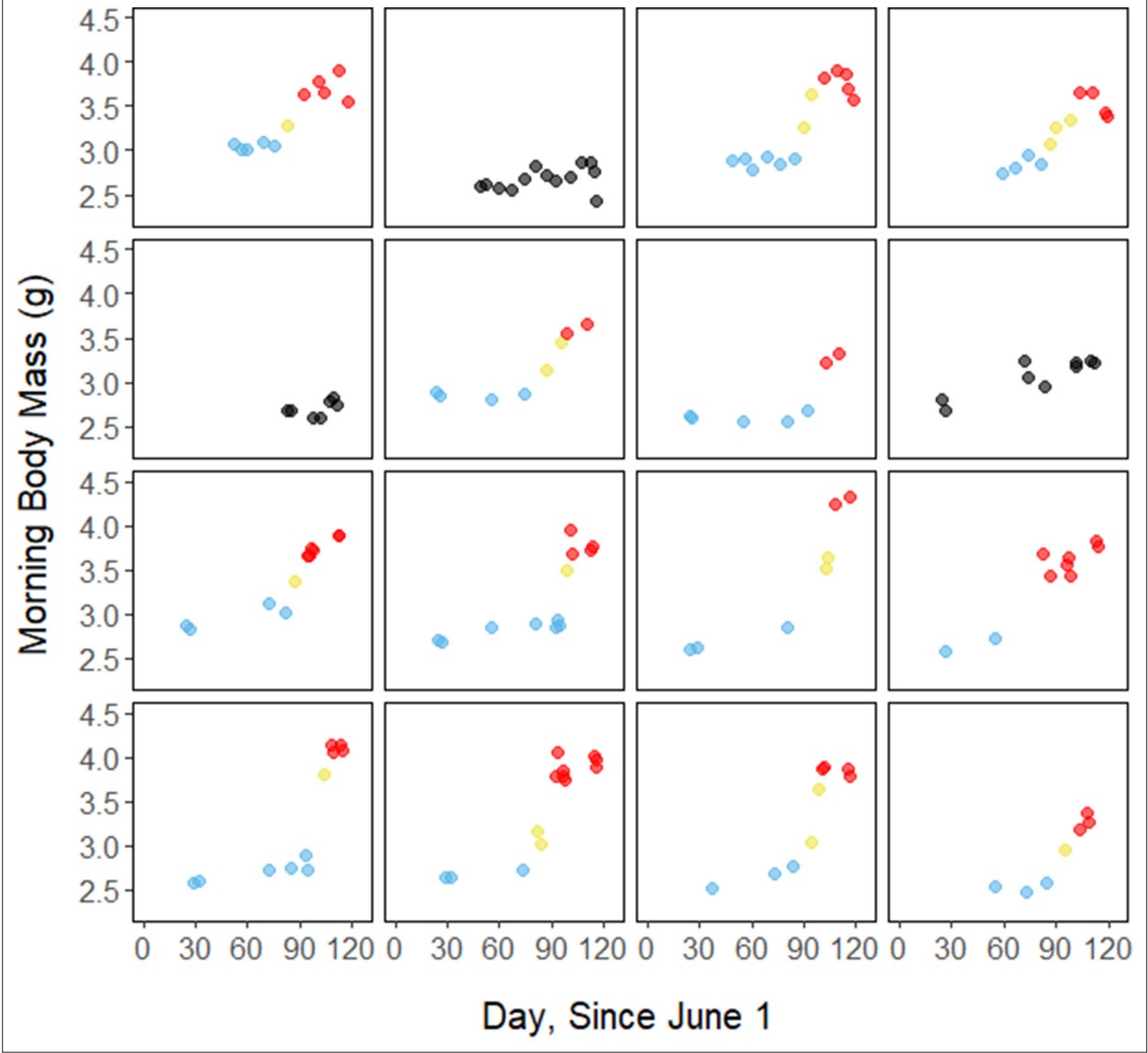

**Figure 1.** Morning body masses following focal observation nights for each individual bird throughout the study period, with points colored by period.

0.39 ± 0.06 kJ·%fat$^{-1}$, p < 0.001, **Appendix 1—figure 3A**), and lost more fat mass overnight (slope: 9.98 ± 1.76 mg$_{fat}$·%fat$^{-1}$, p < 0.001; **Figure 2B**). Additionally, on longer nights, birds spent significantly more energy before entering torpor (slope: 0.40 ± 0.22 kJ·h$_{night}$$^{-1}$, p = 0.041), and lost more fat (slope: 13.5 ± 5.97 mg$_{fat}$·h$_{night}$$^{-1}$, p = 0.014).

Throughout the breeding period, birds consistently entered torpor at a time when they had a relatively low remaining fat level (5.56% ± 0.79%; 142.84 ± 39.17 mg$_{fat}$; **Figure 3B**; **Supplementary file 1**). The fat content at the time of torpor entry did not vary with night length (slope: 0.84 ± 0.78 %$_{fat}$·h$_{night}$$^{-1}$, p = 0.253), or the time of torpor entry (slope: 0.38% ± 0.31% fat·h$_{entry}$$^{-1}$; p = 0.191; **Supplementary file 1**). Consistent with published observations, birds never entered torpor after approximately 75% of the night had elapsed, except on one night in which the bird entered torpor at 80% of the night and aroused within 1 hr (**Hiebert, 1992**; **Figure 3**). On 47 of 55 (85%) bird-nights, the birds either entered torpor when their fat contents reached approximately 5% of body mass, or did not enter torpor if their fat content passed this threshold after 75% of the night had elapsed (**Supplementary file 2**).

## Fattening period

Substantial changes in torpor use accompanied changes in body composition in the late summer when the birds fattened prior to migration. In late August and September, 13 of 16 birds increased body

**Table 1.** Key seasonal differences in morning body mass, fat content at torpor entry, and torpor use with respect to evening fat content and night length.

Signs in parentheses denote the direction of the effect, or where there was no significant relationship this is denoted as 'neither'.

|  | Breeding | Migration |
|---|---|---|
| Morning body mass | Low (2.77 ± 0.05 g) | High (3.73 ± 0.05 g) |
| Fat content at torpor entry | Low, consistent (5.56% ± 0.79%) | High, variable (32.94% ± 0.77%) |
| Torpor propensity | Evening fat content (-) | Evening fat content (+) and night length (+) |
| Torpor entry time | Evening fat content (+) | Neither |
| Torpor duration | Evening fat content (-) | Night length (+) |
| Pre-torpor energy expenditure | Evening fat content (+) and night length (+) | Neither |
| Overnight fat mass loss | Evening fat content (+) and night length (+) | Night length (+) |

mass (slope: $0.02 \pm 0.00$ g·day$^{-1}$, $p < 0.001$; *Supplementary file 1A*) over the span of $10 \pm 1$ days (range: 6–18 days; *Figure 1*; *Supplementary file 1C*). In this short period, birds increased body mass by an average of $0.58 \pm 0.05$ g (range: 0.34–0.90 g). Relative to the start of fattening, birds increased their body mass by an average of $19.6\% \pm 1.6\%$ (range: 11.6–29.0%; *Supplementary file 1C*).

In every period, there was a negative relationship between torpor duration and overnight fat mass loss (slopes: breeding: $-17.23 \pm 0.52$ mg$_{fat}$·hr$^{-1}$, fattening: $-16.82 \pm 1.02$ mg$_{fat}$·hr$^{-1}$, migration: $-15.35 \pm 0.72$ mg$_{fat}$·hr$^{-1}$, $p < 0.001$; *Appendix 1—figure 4*; *Supplementary file 1D*). Because longer torpor durations invariably spared more fat, we predicted that more frequent and longer torpor use, in addition to higher food consumption during the fattening period, would enhance the rate of premigratory fattening. Most interestingly, birds that used torpor for longer on average achieved greater mass gains during the fattening period (slope: $0.07 \pm 0.01$ g·hr$^{-1}$; $p = 0.004$; n = 11; *Figure 4A*; *Supplementary file 1B*). Contrary to our predictions, food consumption did not significantly affect the amount of fattening (slope: $0.00 \pm 0.00$ g·kJ$^{-1}$, $p = 0.996$), and neither mean torpor duration (slope: $-0.60 \pm 0.75$ days·hr$^{-1}$, $p = 0.468$) nor mean food consumption (slope: $-0.27 \pm 0.25$ days·kJ$^{-1}$, $p = 0.343$) was related to the length of the fattening period (*Figure 4*; *Supplementary file 1B*).

## Migration period

In the migration period, birds maintained greater morning body masses compared to the summer ($3.73 \pm 0.05$ g, $p < 0.001$), and the migration period body masses remained stable (slope: $0.00 \pm 0.00$ g·day$^{-1}$, $p = 0.079$; *Figure 1*; *Table 1*; *Supplementary file 1A*). Despite beginning nights with three to five times more fat than they would need to remain normothermic for the entire night at 20°C (mean overnight fat loss on normothermic nights was $194.18 \pm 4.32$ mg$_{fat}$ in the breeding period; $196.02 \pm 4.23$ mg$_{fat}$ during fattening; and $199.22 \pm 4.66$ mg$_{fat}$ during the migration period [pairwise comparisons: $p > 0.745$], when accounting for the effect of night length [$p < 0.001$]) and not approaching the critical threshold apparent in the breeding period, birds used torpor in the migration period at similar frequencies to those of the breeding period (breeding: $61.6\% \pm 11.1\%$; migration: $65.1\% \pm 11.1\%$, $p = 0.946$; *Supplementary file 1D*). However, in stark contrast to the breeding period, birds were more likely to enter torpor on nights when they started with greater fat stores (slope: $0.26 \pm 0.11$; $p = 0.017$) and on longer nights (slope: $2.06 \pm 0.94$, $p = 0.028$; *Appendix 1—figure 2*). Additionally, the time of torpor entry (slope: $-0.08 \pm 0.07$ h$_{entry}$·%fat$^{-1}$; $p = 0.230$), energy expenditure before torpor entry (slope: $-0.06 \pm 0.05$ kJ·%fat$^{-1}$; $p = 0.204$; *Figure 3A*), and overnight fat mass loss (slope: $-1.54 \pm 1.24$ mg$_{fat}$·%fat$^{-1}$; $p = 0.148$; *Figure 2B*) were not significantly related to evening fat content. Time of torpor entry relative to the start of the night (slope: $-0.51 \pm 0.56$ h$_{entry}$·h$_{night}^{-1}$, $p = 0.333$), energy expenditure before torpor entry (slope: $-0.41 \pm 0.41$ kJ·h$_{night}^{-1}$, $p = 0.276$), and overnight fat mass loss (slope: $-0.26 \pm 10.83$ mg$_{fat}$·h$_{night}^{-1}$, $p = 0.979$) were consistent across the migration period and not significantly related to night length (*Supplementary file 1D*). Conversely, torpor duration was not significantly related to evening fat stores (slope: $0.10 \pm 0.07$ hr·%fat$^{-1}$; $p = 0.124$;

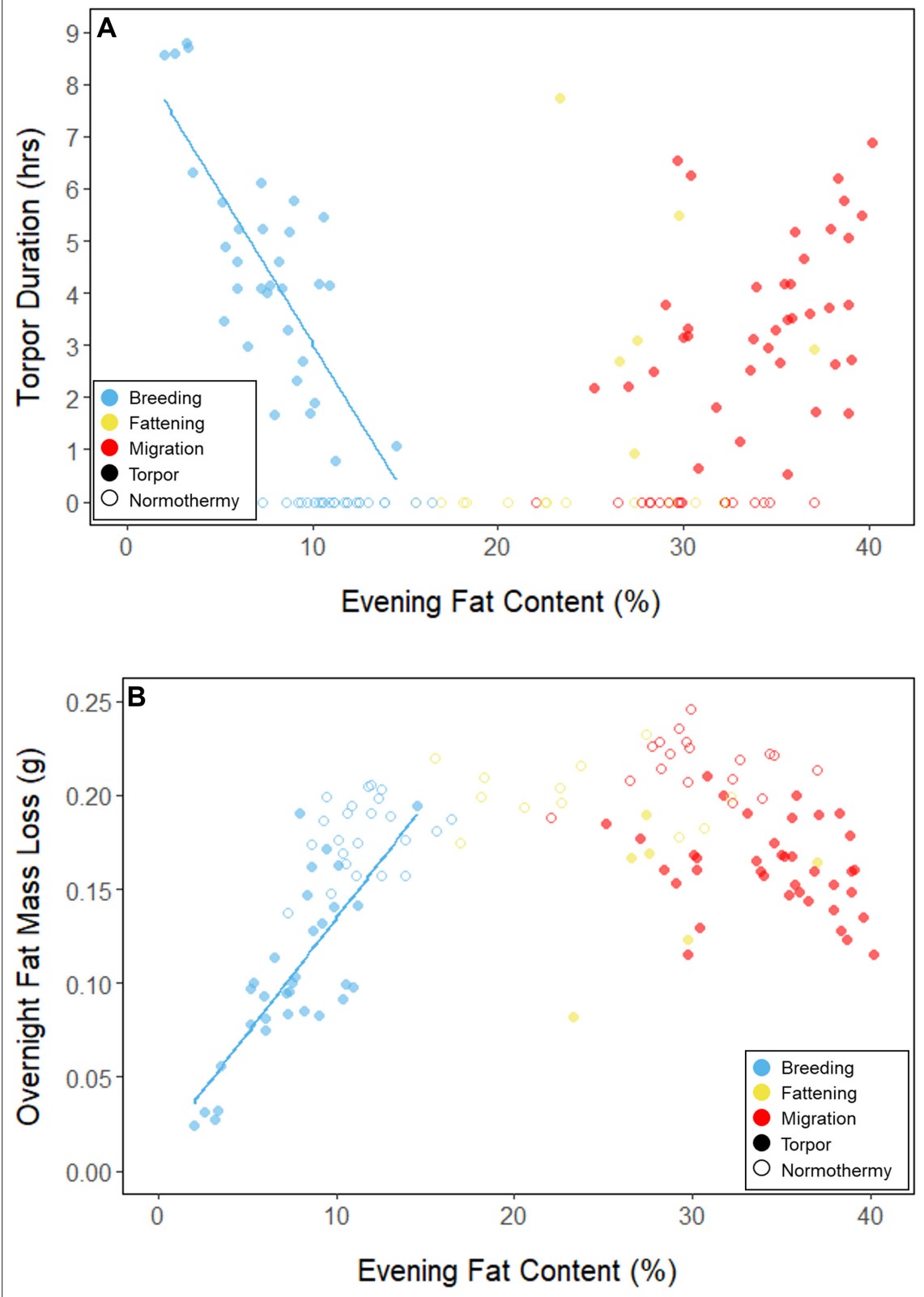

**Figure 2.** Relationships between evening fat content and (**A**) torpor duration, and (**B**) overnight fat mass loss, within each period, with points and significant trendlines colored by period and shaped by torpor use.

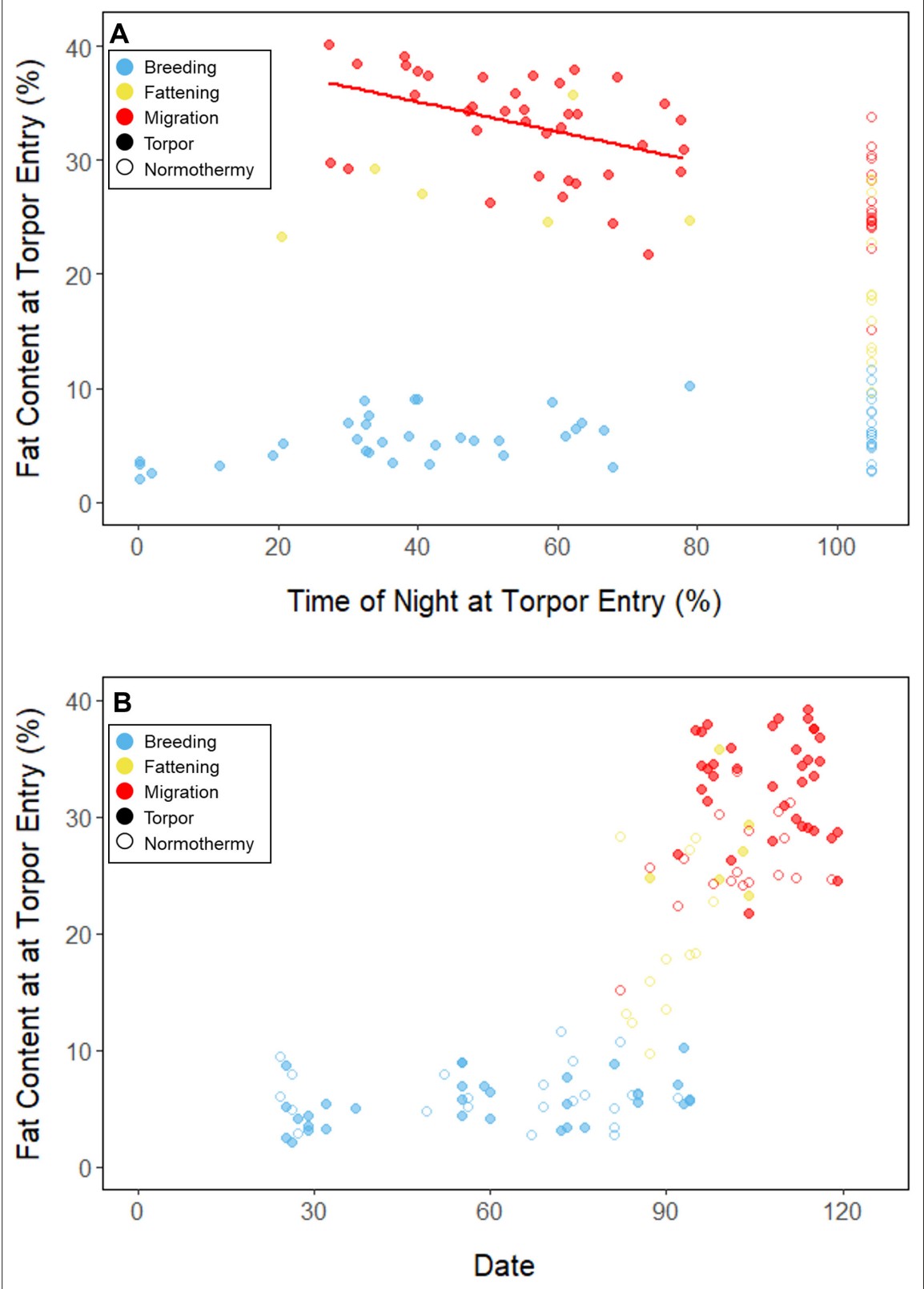

**Figure 3.** Relationship between fat content at the time of torpor entry (%) and (**A**) the time of torpor entry (as % of night), and (**B**) date, with points and significant trendlines colored by period and shaped by torpor use.

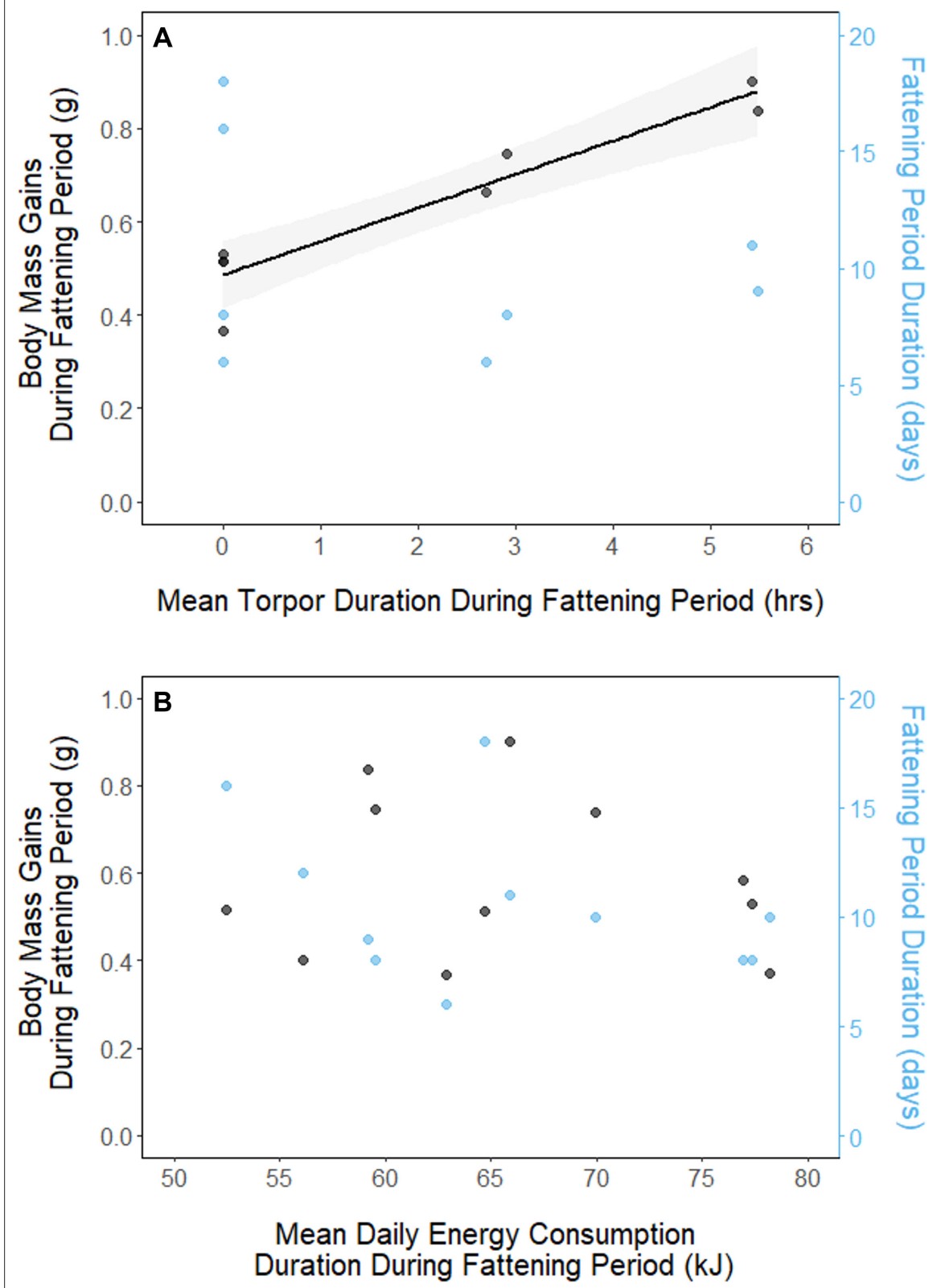

**Figure 4.** Relationships between magnitude of increases in body mass (black, left axis) and fattening period duration (blue, right axis), and (**A**) mean torpor duration, and (**B**) mean daily energy consumption, within the fattening period. Trendlines are shown for significant slopes and the shaded area represents 95% confidence intervals.

*Figure 2A*) but was positively correlated with night length (slope $1.18 \pm 0.58$ $h_{torpor} \cdot h_{night}^{-1}$, p = 0.031). In contrast to the breeding period, average fat content at the time of torpor entry was substantially greater in the migration period ($32.94\% \pm 0.77\%$; $1249.08 \pm 38.34$ $mg_{fat}$; p < 0.001; *Figure 3B*; *Supplementary file 1D*). Additionally, while there was a significant negative relationship between time of torpor entry and the fat content at that time (slope: $-1.20 \pm 0.35$ $\%$ $fat \cdot h_{entry}^{-1}$, p < 0.001; *Figure 3A*; *Supplementary file 1D*), fat content at torpor entry did not vary with night length (slope: $0.99 \pm 1.41 \%_{fat} \cdot h_{night}^{-1}$, p = 0.450).

## Discussion

Ruby-throated hummingbirds exhibit substantial shifts in body composition and use of torpor between the summer breeding period and the migration period as a means to achieve premigratory fat gains (*Table 1*). In the summer, torpor is sensitive to critically low endogenous energy reserves; however, when the birds fatten for migration, this rule is abandoned, and the birds enter torpor with high levels of fat. While it does not appear that torpor initiation in the migration period is simply sensitive to a high energy threshold, the abandonment of the emergency threshold indicates that the rules governing torpor use are dependent on life history stage, and that hummingbirds may employ torpor as part of various energy management strategies throughout the annual cycle.

In the summer breeding period, ruby-throated hummingbirds reserve torpor for times when they face critically low fat stores during the night. At air temperatures of 20°C, which free-living birds commonly experience, torpor initiation is primarily driven by instantaneous fat stores. On nights when they started with lower evening fat stores birds depleted energy stores to critically low levels earlier in the night, and thus remained torpid longer, irrespective of night length. These results support the hypotheses that torpor use is sensitive to a low, consistent threshold of fat during the breeding period, and that torpor is an energy emergency survival strategy mechanism that protects hummingbirds from depleting energy stores during the night or before they can reach their first meal in the morning (*Hainsworth et al., 1977*; *Hiebert, 1992*; *Powers et al., 2003*).

The birds obeyed the average threshold on the clear majority of focal nights (47 out of 55 nights). On eight nights, 6 of the 13 fattener birds either crossed the threshold and did not enter torpor, or entered torpor without ever crossing the threshold, but we argue it is not surprising that there were a few exceptions to the threshold rule. The threshold could occasionally be overridden to remain normothermic despite reaching critical energy levels if extended bouts of torpor on subsequent nights impede regenerative processes associated with normothermy and sleep, or increase risk of predation (*Bouma et al., 2010*; *Carr and Lima, 2013*; *Hiebert, 1990*). Similarly, we and other researchers have observed hummingbirds prematurely rewarming after inadvertent light or noise disturbances, which suggests that the threshold override and emergency arousal mechanisms could be related (A. Shankar, E. Eberts, personal observations). Although we did not observe any obvious signs of greater stress in some individuals than others, it is also possible that some birds (e.g. those that were more recently captured) were more sensitive to handling and confinement in the respirometry chamber and therefore initiated torpor despite not having depleted fat levels to the threshold level (*Hiebert et al., 2000*). Furthermore, although temperature, precipitation, and food availability were controlled in this study, these factors would likely affect free-living hummingbirds' torpor use decisions by effecting the amount of fat the birds start each night with and the rate that they expend that fat (*Hainsworth et al., 1977*; *McGuire et al., 2021*). For example, the birds were fasted to ensure accurate body composition measurements, but if the birds were allowed to eat during the last 2 hr of the day, it is likely that they would have amassed larger evening fat stores and therefore would have entered torpor later in the night or not at all (*Eberts et al., 2019*). Additionally, the level of the energy-store threshold could be modulated in response to anticipated energy demand. For instance, free-living birds at our study site during the breeding period experienced minimum nighttime air temperatures between 10°C and 25°C (London International Airport, Ontario, Canada). At colder temperatures, normothermic hummingbirds would need to sustain a higher resting metabolic rate and may therefore anticipate an energy emergency earlier in the night (*Hiebert, 1990*). While these factors likely play a role in the complex decision-making that governs free-living hummingbird torpor use during the breeding season, our evidence strongly supports the existence of an 'adipostat' mechanism that initiates compensatory physiological changes depending on the instantaneous level of endogenous energy stores (*Boyer and Barnes, 1999*; *Powers et al., 2003*).

The patterns and role of torpor use drastically shifts as hummingbirds accumulate fat stores prior to migration. Contrary to the breeding period, torpor propensity and torpor duration were primarily driven by night length rather than evening fat content, and torpor entry, energy expenditure before torpor entry, and overnight fat loss did not vary with night length. Instead of initiating torpor to survive the night whenever they reach a critically low threshold, hummingbirds appear to enter torpor after a consistent amount of time, so that they expend a predictable amount of energy and achieve greater energy savings as they experience progressively longer nights in the late summer. This suggests that during the migration period, birds maximize their time in torpor, but must remain normothermic for a consistent period of time, perhaps accounting for the time it takes to process their final evening meal and their blood sugar declines to a level allowing for torpor initiation (*Eberts et al., 2019*). This normothermic period could also allow the birds to achieve sufficient sleep before entering torpor when the regenerative and immunological benefits of sleep are unlikely to occur (*Bouma et al., 2010*). Overall, the seasonal change in torpor use, from a survival strategy initiated by an 'adipostat' mechanism to a more routine use of torpor to maximize energy savings and build fat stores prior to migration, shows that torpor is a critical energy management strategy that allows migratory hummingbirds to balance fuel supply and demand during a particular season. Similar torpor use patterns have been well studied in mammals, though studies investigating avian endocrine mechanisms are needed to elucidate the proximate factors of torpor initiation across taxa.

The similarities between the life histories of North American hummingbirds and bats provide an important lens through which to interrogate the ultimate drivers of torpor use throughout the annual cycle (*McGuire et al., 2012*). Using torpor while roosting could allow both hummingbirds and bats animals to maintain high fuel stores when feeding opportunities are constrained by migratory priorities. Unlike most nocturnal migrant birds that can replenish fuel stores during the day, ruby-throated hummingbirds are diurnal migrants that do not forage at night. In an opposite but parallel manner, North American bats migrate at night when they would otherwise forage and do not forage during the day. In both of these types of animals, foraging, migrating, and sleeping can be mutually exclusive endeavors, but torpor can allow them to conserve energy and avoid making extended refueling stopovers (*McGuire et al., 2014*). While we did not study free-living birds while they migrated, the drastic shift in the relationship between hummingbird body composition and torpor use, and of the link between torpor duration and mass gains in the late summer compellingly support the 'torpor-assisted migration' hypothesis (*Carpenter and Hixon, 1988*; *Hiebert, 1993*; *Hou and Welch, 2016*; *McGuire et al., 2012*).

The magnitude and timing of fattening in our captive birds resembled those documented in some free-living ruby-throated hummingbirds (present study: 0.58 ± 0.05 g, 19.6% ± 1.6% over 10 ± 1 days; *Hou and Welch, 2016*: ~ 0.65 g, or 17% over 4 days). However, not all birds in this study showed such substantial body mass changes; there was continuous variation in magnitude and duration of fattening within the fattener birds, and three non-fattener birds maintained relatively lean body compositions throughout the study period. The ranges in magnitude and timing of fattening are not surprising because hummingbirds are asocial birds that do not migrate in flocks, and we would not expect them to have finely synchronized timing of migratory preparation, especially in the absence of natural ecological cues. Furthermore, our captive birds varied in wing feather wear and activity level, suggesting that the amount of fat they could deposit was limited by how much extra weight they could carry, determined by wing morphology, pectoral mass, and power output (*Chai, 1997*; *Dakin et al., 2020*). These interindividual differences could also reflect disparate energy management strategies observed among free-living individuals. For example, a study examining migratory paths of juveniles suggested that adults take a more direct route than individuals (*Zenzal and Moore, 2016*). These disparate migratory strategies could be driven by differences in competitive abilities or experience from previous migratory journeys (*Carpenter et al., 1993b*; *Carpenter et al., 1993a*; *Hou and Welch, 2016*; *Kodric-Brown and Brown, 1978*; *Welch et al., 2008*). This hypothesis is supported by the fact that one of the three non-fattener birds was the only juvenile in our study, and ongoing work is investigating potential age/sex class differences in an ecologically relevant context.

This is the first study to non-terminally and repeatedly sample the body composition of individuals to accurately define a consistent rule governing torpor use in hummingbirds: birds will enter torpor when their fat stores reach a consistently low fat threshold (5%), during life history stages where a relatively lean body composition is advantageous. This rule may explain the typically low degree of torpor

use in dominant individuals and large species, and the more frequent use of torpor in subordinate individuals and smaller species (*Powers et al., 2003*). Moreover, our study is consistent with the long standing, but heretofore unproven, assumptions of the key role of torpor in premigratory fattening and in refueling at migratory stopover sites. Hummingbirds dramatically shift their rules for torpor use and enter torpor at high fat levels during times when it is advantageous to accumulate excess energy stores. These findings demonstrate the versatility of torpor as an energy management mechanism throughout the annual cycle and have important implications for understanding the physiological basis of torpor initiation.

## Materials and methods

### Study animals

Adult (and one juvenile) male ruby-throated hummingbirds (*A. colubris*; n = 16; capture mass: 2.54–3.2 g) were captured with a modified box trap (drop door trap) in London, ON, Canada, at the University of Western Ontario. Captive hummingbirds were housed individually in EuroCage enclosures (Corners Ltd, Kalamazoo, MI), measuring 91.5 cm W × 53.7 cm H × 50.8 cm D, at the University of Western Ontario's Advanced Facility for Avian Research. Once captive, birds were fed ad libitum on a 20% (*w/v*) solution of a Nektar-Plus (Guenter Enderle, Tarpon Springs, FL), and were housed at 20°C and approximately 50% relative humidity. Birds experienced semi-natural photoperiods that were changed approximately weekly, ranging from 15 hr light/9 hr dark to 12 hr light/12 hr dark. These photoperiods are reflective of the birds' natural summer photoperiod, as the data were collected in the summers 2018 and 2019. Lights were abruptly turned on in the morning and shut off in the evening. The birds transitioned from a breeding condition in the beginning of the study period to a migratory condition in end of the study period. Details of animal husbandry and all experiments were approved by the University of Toronto (protocol # 20011649) and the University of Western Ontario Animal Care Committees (protocol #2018–092). Hummingbirds were captured under Ontario Collecting Permit SC-00041.

### Body composition

This study uses QMR to measure body composition. QMR is a technology developed in the last 15 years that allows for non-invasive measurement of the masses of fat, lean tissue, and total body water (*Guglielmo et al., 2011*). QMR allows for short scan times (2–3 min), high precision and accuracy, and the ability to measure resting, non-anesthetized animals (*Guglielmo et al., 2011*). We used an Echo-MRI (Echo Medical Systems, Houston, TX) with an A10 antenna for measuring birds < 10 g. We calibrated the QMR machine with 1.5 g canola oil and 10 g water standards and scanned these standards daily to check the calibration; scans were set at three accumulations. On focal nights, birds were scanned three to five times in the evening and the morning; the means of the values from these scans were calculated. QMR, paired with respirometry allows us to non-invasively and accurately estimate the level of endogenous energy stores throughout the night, and specifically at the time of torpor initiation.

### Respirometry

This study uses respirometry to calculate rates of energy expenditure. Oxygen consumption and carbon dioxide production rates overnight were obtained via push-flow respirometry using an FC-1B oxygen analyzer, a CA-2A carbon dioxide analyzer (Sable Systems International, Las Vegas, NV). Air was first passed through a dew point generator set at 10–15°C and then was flowed into the chambers through Bev-a-line tubing at a rate of 150 mL/min. The excurrent airstream was subsampled at 50 mL/min. Subsampled air first passed through a water vapor meter, which measured water vapor pressure (kPa) (RH-300; Sable Systems International). The air then passed through a column containing indicating Drierite (W.A. Hammond DRIERITE, Xenia, OH), the carbon dioxide gas analyzer, and the oxygen analyzer. Analogue voltage outputs from the thermoresistor, oxygen, and carbon dioxide analyzers, flow meter, water vapor pressure, and in-line barometric pressure sensors were recorded at 1 s intervals over the duration of the trial (9–12 hr) using EXPEDATA software (v.1.9.27; Sable Systems International) and were recorded on a laptop computer.

Raw data were corrected to standard temperature and pressure, and rates of oxygen consumption ($V_{o_2}$) and carbon dioxide production ($V_{co_2}$) were calculated in Expedata using standard equations (12; equations 10.6, 10.7, respectively). The rate of oxygen consumption ($V_{o_2}$), the respiratory exchange ratios (RER = $V_{o_2}$ / $V_{co_2}$, indicates primary metabolic fuel type), and the oxyjoule equivalent were used to calculate the rate of energy expenditure (kJ·min⁻¹) ($E_{rate}$=(16 + 5.164*RER)* $V_{o_2}$) (*Lighton, 2008*). Where RER was extraneously below 0.71 or above 1.0 it was bound at these limits to satisfy the assumptions of this equation. Total nighttime and metabolic state-specific energy expenditures were calculated by integrating the metabolic rate ($E_{rate}$) over time.

## Experimental protocol

In each overnight experiment, birds were food-deprived for approximately 2 hr (1.66 ± 0.44 hr) prior to lights-off to ensure that crop stores were emptied (for accurate body composition measurements) and that the only available sources of energy were endogenous fat and lean mass. Body composition was measured using QMR before the birds were placed in respirometry chambers (10 cm W × 10 cm H × 20 cm D) at 20°C and the lights were turned off. Torpid birds could not be scanned because disturbing them would cause them to unnaturally arouse and because their cooler body temperatures could decrease scan precision and consistency (*Guglielmo et al., 2011*). Immediately following lights-on in the morning, body composition was measured. Air temperature was measured directly outside the chamber via a thermoresistor; although air temperature inside the chamber was not recorded, experiments show the inside and outside air temperatures were within 1°C.

## Data processing

Between June and September 2018 and 2019, we recorded repeated overnight measurements on 15 adult male birds and 1 male juvenile, for a total of 158 bird-nights. The birds began exhibiting increased body masses (indicative of migratory condition) at different times in the late summer (mean: August 28; range: August 12–September 8). In order to identify periods of distinct rates of change in body mass, we analyzed the rate of change in morning body mass across the study period (*Supplementary file 3*). We first smoothed the morning body mass trace with a smoothing parameter (0.35) that we identified through an iterative process. We calculated the first derivative of this smoothed body mass trace, and defined periods based on bird-specific 'cut-off' slopes that we calculated as 75% of the maximum slope (g·day⁻¹). We defined 'breeding' as periods where the rate of change was less than the cut-off slope, 'fattening' as periods where the rate of change was greater than the cut-off slope, and 'migration' as periods where the rate of change was less than the cut-off slope and after the start of the fattening period. When this analysis yielded periods that clearly disagreed with visual inspection of the curve, we slightly adjusted the bounds of the fattening period to fit a more realistic pattern. Three birds that did not fatten were categorized as 'non-fatteners' and were excluded from the statistical models that regard seasonal changes in torpor use.

We calculated torpor propensity as the percentage of nights the birds entered torpor of the total number of observation nights for each bird within each period. In order to calculate the energetic characteristics of torpor, such as the fat content at the time of torpor entry and duration, the temporal characteristics of torpor must be clearly defined. In much of the avian and mammalian torpor literature, torpor entry is defined by phrases such as when the metabolic rate 'abruptly declines', or by criteria such as a threshold value of a set proportion of the average normothermic resting values of body temperature or metabolic rate (*Ruf and Geiser, 2015*; *Shankar et al., 2020*; *Wolf et al., 2020*). However, these various and often vague definitions are problematic for repeatability and our understanding of energy metabolism at specific stages of torpor. In order to identify accurate and repeatable periods of consistent metabolic states, we analyzed the rate of change in $V_{o_2}$ across the night (*Appendix 1—figure 6*). We first linearly interpolated $V_{o_2}$ and smoothed this trace using a smoothing parameter (0.6) that we identified through an iterative process. We calculated separate smoothed traces for the time before (containing entry) and after (containing arousal) the end of torpor/start of arousal. To determine this intermediate point we calculated the derivative of a smoothed trace of the entire night (using night-specific smoothing factor), and preliminarily identified the approximate end of torpor/start of arousal as the minute the rate of change was greater than an arousal cut-off slope of 0.005 $V_{o_2}$ ·min⁻¹. We then calculated the first derivative of each of the entry and arousal smoothed $V_{o_2}$ traces, and defined states based on entry and arousal cut-off slopes. We defined 'entry' as periods

where the rate of change was greater than two times the standard deviation of the mean rate of change during the normothermic period before torpor (which was identified by analyzing the preliminary smoothed curve where the rate of change was less than $-0.003\ V_{O_2} \cdot min^{-1}$). We defined 'arousal' as periods where the rate of change was greater than five times the standard deviation of the mean rate of change during the torpid period; we defined the end of the arousal period as the point when the bird exhibited peak $V_{O_2}$. We defined 'torpor' as periods where the $V_{O_2}$ was stable and between entry and arousal. We also annotated points before the start of entry and after the end of arousal as 'normo-pre' and 'normo-post', respectively. This process yielded accurate and repeatable metabolic state annotations of each minute.

The initial evening and the final morning percent fat content were calculated as fat mass (g)/body mass (g). We used the rate of energy expenditure, cumulative energy expenditure, and initial fat mass measurements to estimate instantaneous percent fat content throughout each night. These data allowed for the estimation of the amount of energy reserves, relative to body mass, of each bird at the time of torpor initiation. We calculated this value by subtracting the fat mass equivalent (1 $g_{fat}$/37 kJ) of cumulative energy expenditure at torpor entry from evening fat mass and dividing the result by morning body mass. We calculated torpor duration (hr) as the time between the start of torpor entry until the beginning of arousal (excluding arousal). We calculated overnight mass losses as the fat mass equivalent of the amount of overnight energy expenditure. Lastly, we calculated body mass increases as the change in body mass from the beginning to the end of the fattening period, relative to the smoothed morning body mass trace used to determine the periods.

## Statistical analyses

We used mixed effects analyses to evaluate the relationships between various response and predictor variables, while accounting for intra-individual differences (*Bates et al., 2015*). First, we used linear mixed effects analyses to determine how body mass, fat content, lean mass, and food consumption changed within and among (*with respect to* date) each period of distinct mass change (with day length as a covariate for the food consumption model). We evaluated the overall relationships between body mass and fat mass, and body mass and lean mass, using a repeated measures correlation test ('*rmcorr*' R package; *Bakdash and Marusich, 2017*).

We used a linear mixed effects model to compare mean torpor propensity among periods, and a logistic mixed effects model to compare the influence of evening fat content on probability of occurrence of torpor within each period. We also used linear mixed effects models to determine how torpor duration, energy expenditure before torpor entry, and overnight fat mass loss changed with respect to evening fat content within and among periods, and with respect to night length within and among periods. We used linear mixed effects models to determine how % fat at the time of torpor entry changed with respect to date and the time of night, with night length as a covariate, within and among periods. We also used linear mixed effects models to determine how overnight mass loss changed with respect to torpor duration within and among periods. Lastly, we used linear models to evaluate the effect of mean torpor duration and mean daily food consumption within the fattening period on the magnitude and duration of the fattening period.

We performed all statistical analyses using *R Development Core Team, 2020*. To generate linear and logistic mixed effects models, we used the '*lme4*' package, with Bird ID as a random effect (*Bates et al., 2015*). For linear models we used the '*lm*' function. For each response variable, we iteratively compared several combinations of relevant fixed effects, and used the AICs to determine the most parsimonious model. The model with the lowest AIC by at least two points was considered the best, and we verified that residuals of this model showed homoscedasticity and normality. We determined p-values using the '*anova*' function on each model generated using the '*lmerTest*' package, and made pair-wise comparisons within and among periods using the '*emmeans*' package. All values are given as estimated marginal means ± standard error, unless otherwise indicated, and significance was taken at α < 0.05.

## Acknowledgements

We thank AFAR staff (Francis Boon and Michela Rebuli), and undergraduate assistants. We also thank Welch lab members (University of Toronto), Morag Dick, and Anusha Shankar for discussions. We also thank Alex Gerson and his lab members for providing housing, and for discussions. This work was

supported by a Natural Sciences and Engineering Research Council of Canada (NSERC) Discovery Grant 06129–2015 RGPIN and Human Frontier Science Program (number RGP0062/2016) to KCW; and NSERC Discovery Grant 05245–2015 RGPIN, and Canada Foundation for Innovation, Ontario Research Fund to CGG.

## Additional information

### Funding

| Funder | Grant reference number | Author |
|---|---|---|
| Natural Sciences and Engineering Research Council of Canada | 06129-2015 RGPIN | Kenneth C Welch Jr |
| Human Frontier Science Program | RGP0062/2016 | Kenneth C Welch Jr |
| Natural Sciences and Engineering Research Council of Canada | Discovery Grant 05245-2015 RGPIN | Christopher G Guglielmo |
| Canada Foundation for Innovation | Ontario Research Fund | Christopher G Guglielmo |

The funders had no role in study design, data collection and interpretation, or the decision to submit the work for publication.

### Author contributions
Erich R Eberts, Conceptualization, Data curation, Formal analysis, Investigation, Methodology, Project administration, Software, Visualization, Writing – original draft, Writing – review and editing; Christopher G Guglielmo, Conceptualization, Formal analysis, Methodology, Resources, Supervision, Writing – review and editing; Kenneth C Welch Jr, Conceptualization, Formal analysis, Funding acquisition, Methodology, Resources, Supervision, Writing – review and editing

### Author ORCIDs
Erich R Eberts  http://orcid.org/0000-0003-4259-8752
Kenneth C Welch Jr,  http://orcid.org/0000-0002-3283-6510

### Ethics
Details of animal husbandry and all experiments were approved by the University of Toronto (protocol # 20011649) and the University of Western Ontario Animal Care Committees (protocol #2018-092). Hummingbirds were captured under Ontario Collecting Permit SC-00041.

### Decision letter and Author response
Decision letter https://doi.org/10.7554/eLife.70062.sa1
Author response https://doi.org/10.7554/eLife.70062.sa2

## Additional files

### Supplementary files
• Transparent reporting form

• Supplementary file 1. Supplementary results tables. (A) Estimated marginal means and slopes of body composition and food consumption variables, descriptions of the statistical models, and statistical comparisons within and among periods. (B) Absolute and relative increases in body mass, fat mass, and fat content during the fattening period. (C) Correlations between body mass gains and duration of the fattening period, with mean torpor duration in the fattening period and food consumption in the fattening period. (D) Estimated marginal means and slopes of torpor use variables, descriptions of the statistical models, and statistical comparisons within and among periods.

• Supplementary file 2. Instantaneous percent fat content over time (as % of night) throughout each

summer, fattening, and migration period, as well as during the whole study period for non-fatteners. Red lines represent normothermic nights and blue lines represent torpid nights. The average breeding threshold ±1 standard error is indicated by horizontal dashed black and greay lines, respectively.

• Supplementary file 3. Morning body mass (black points) following focal nights across the entire study period, starting at the date of capture. These data points were smoothed (greay line), and the slope of these points was used to define breeding, fattening, and migration periods for each bird, which are shaded blue, yellow, and red, respectively. Non-fatteners are also included and shaded dark red. The first panel shows night length throughout the study period.

## Data availability

All data is available in the main text or the supplementary materials. Analyses reported in this article can be reproduced using the data and code provided by Eberts et al., 2021.

The following dataset was generated:

| Author(s) | Year | Dataset title | Dataset URL | Database and Identifier |
|---|---|---|---|---|
| Eberts E, Guglielmo C, Welch Jr K | 2021 | Data from: Reversal of the adipostat control of torpor during migration in hummingbirds | https://doi.org/10.5061/dryad.p8cz8w9qg | Dryad Digital Repository, 10.5061/dryad.p8cz8w9qg |

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

## Appendix 1

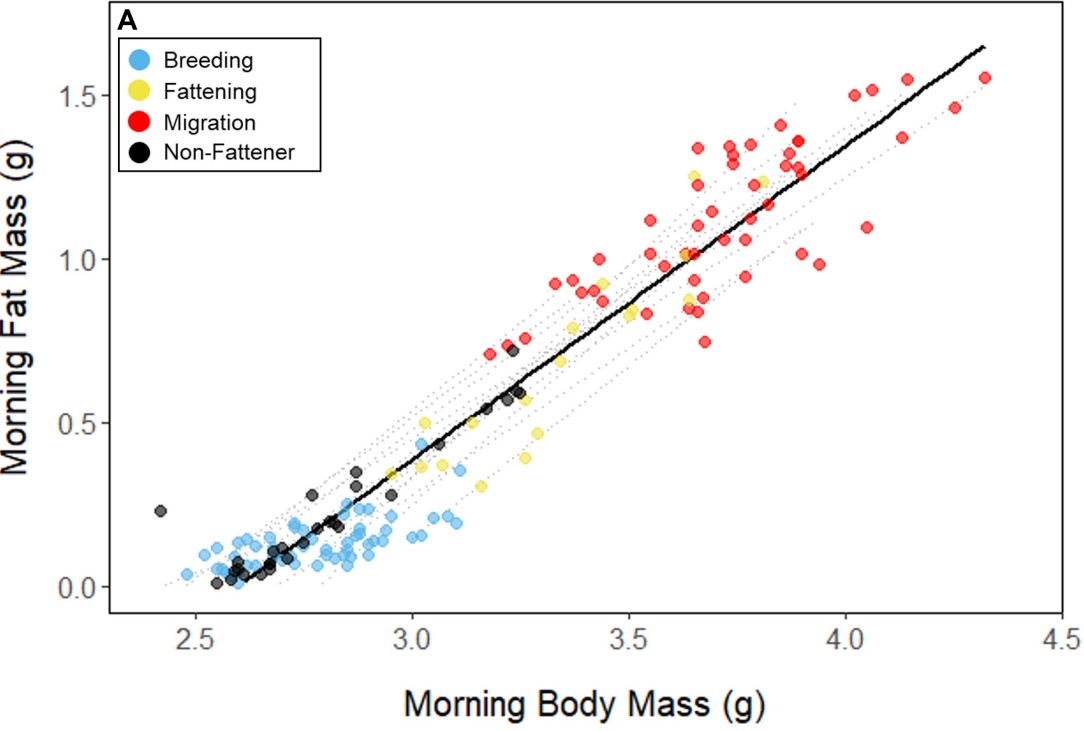

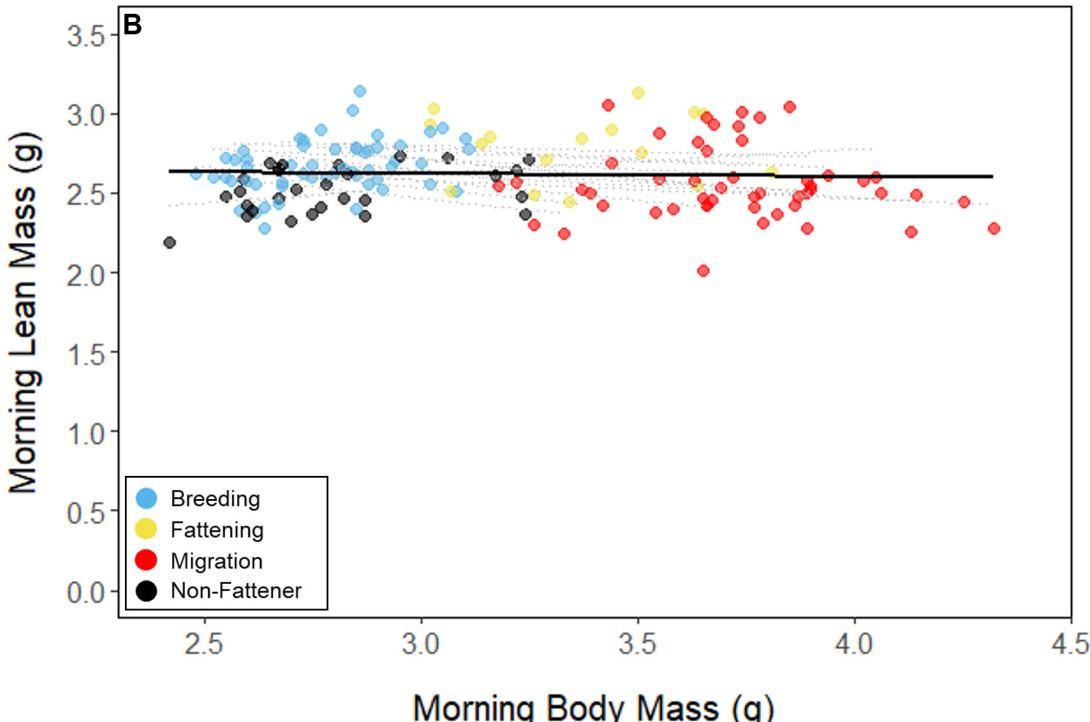

**Appendix 1—figure 1.** Relationships between body mass and (**A**) fat mass, and (**B**) lean mass, with points colored by period. Black lines represent the overall linear relationship during the entire study period and dashed lines represent bird-specific regressions.

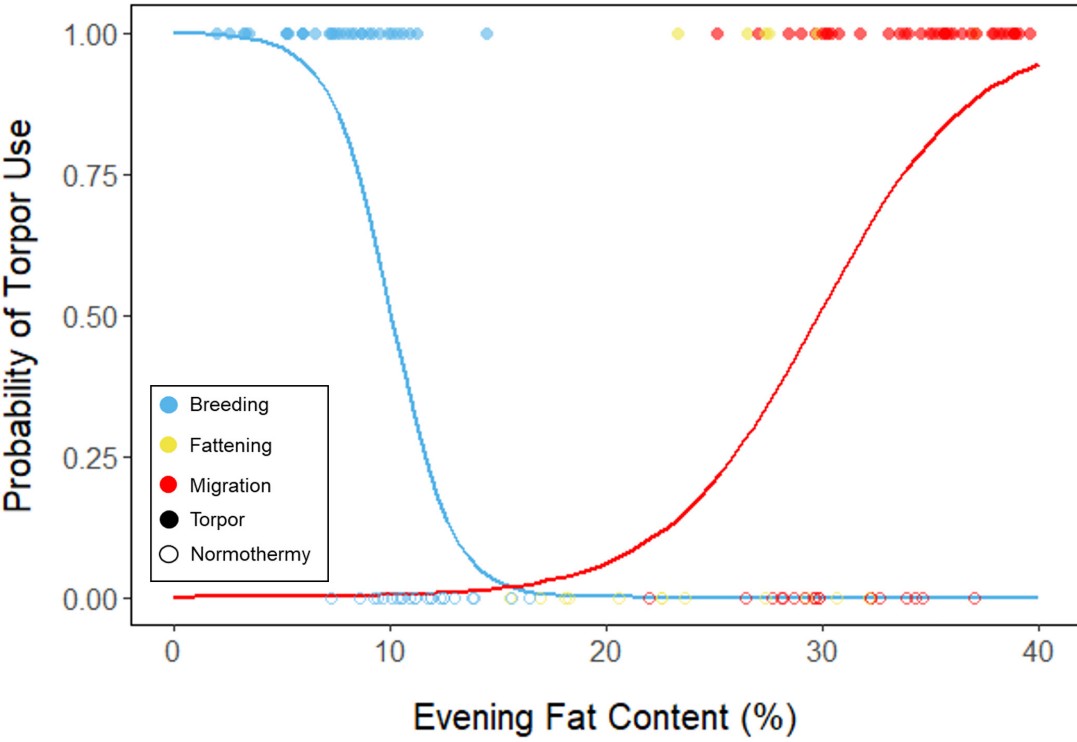

**Appendix 1—figure 2.** Logistic regression of torpor use as a function of evening fat content, with points and significant trendlines colored by period and shaped by torpor use. Lines are predicted logistic curves estimating the probability of torpor use with respect to evening fat mass.

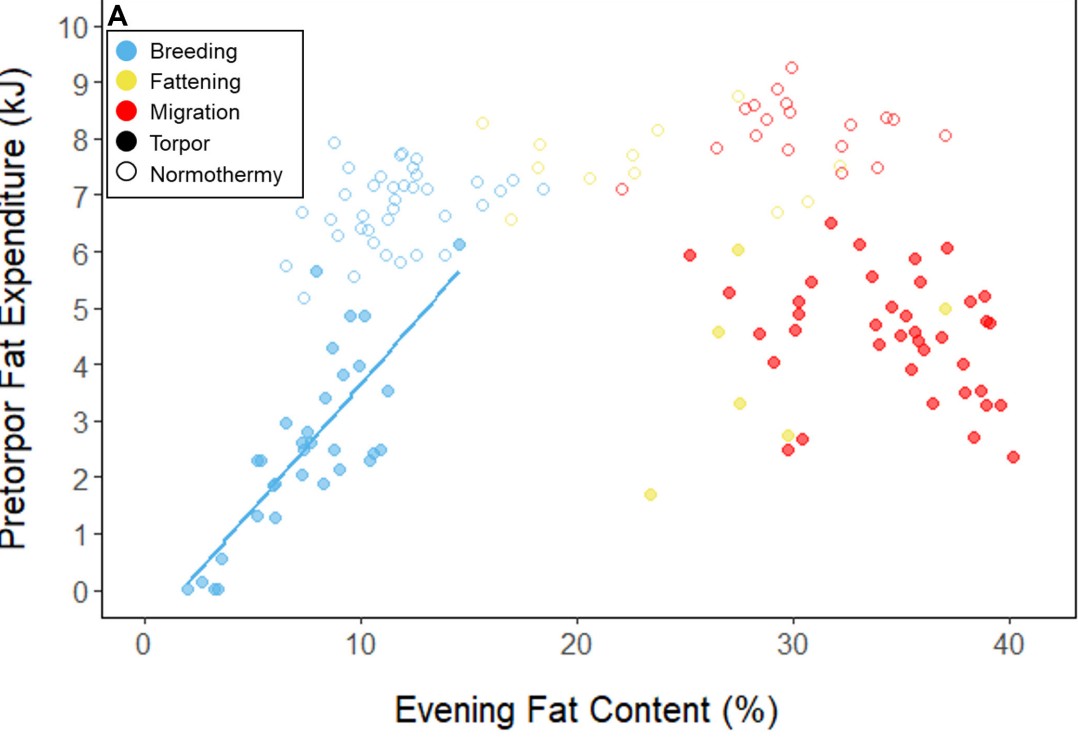

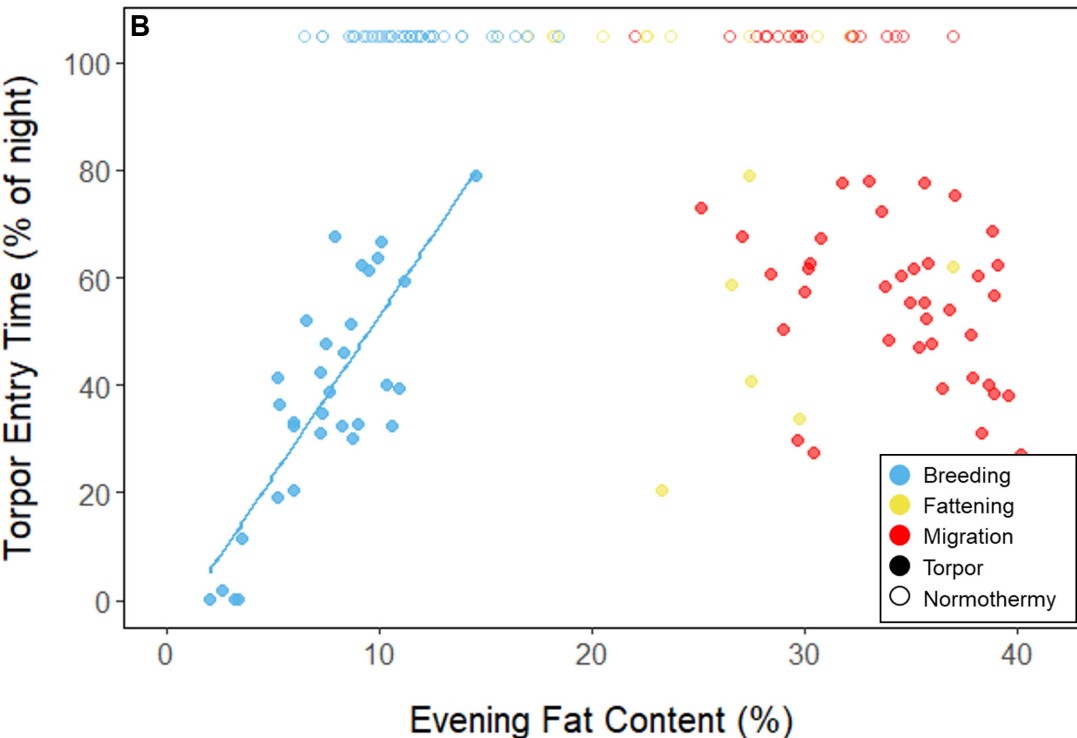

**Appendix 1—figure 3.** Relationships between evening fat content and (**A**) energy expenditure before torpor entry, and (**B**) time of torpor entry, within each period, with points and significant trendlines colored by period and shaped by torpor use.

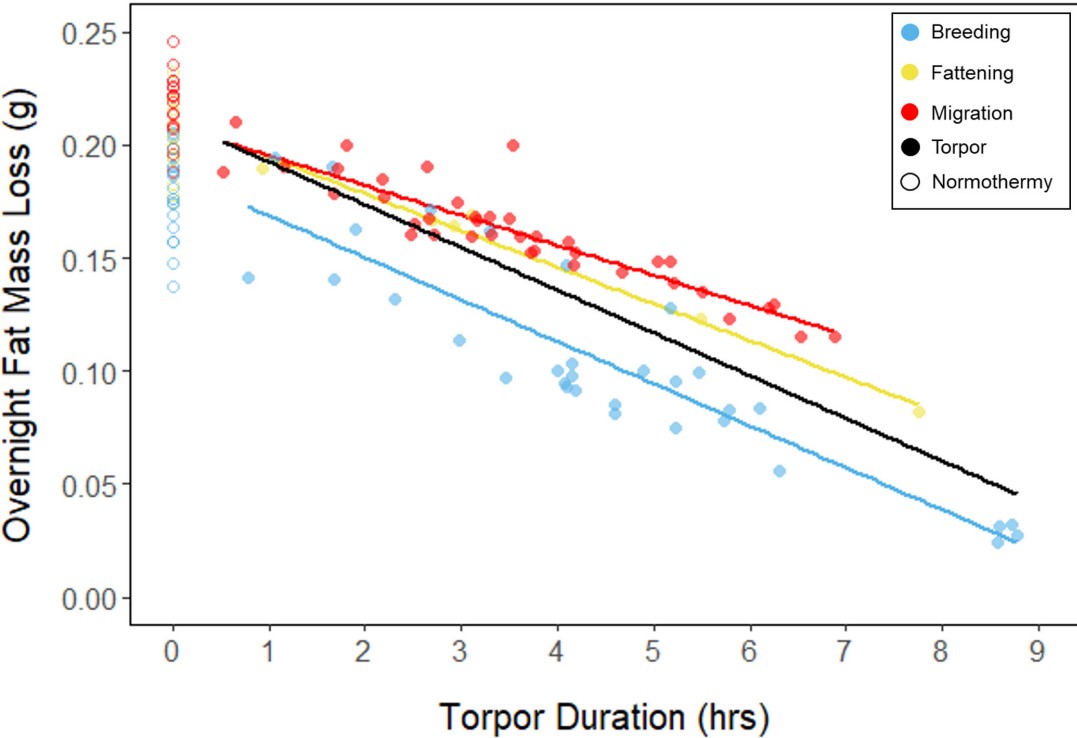

**Appendix 1—figure 4.** Relationships between torpor duration and overnight fat mass loss, within each period, with points and significant trendlines colored by period and shaped by torpor use. Black line indicates overall linear relationship irrespective of period.

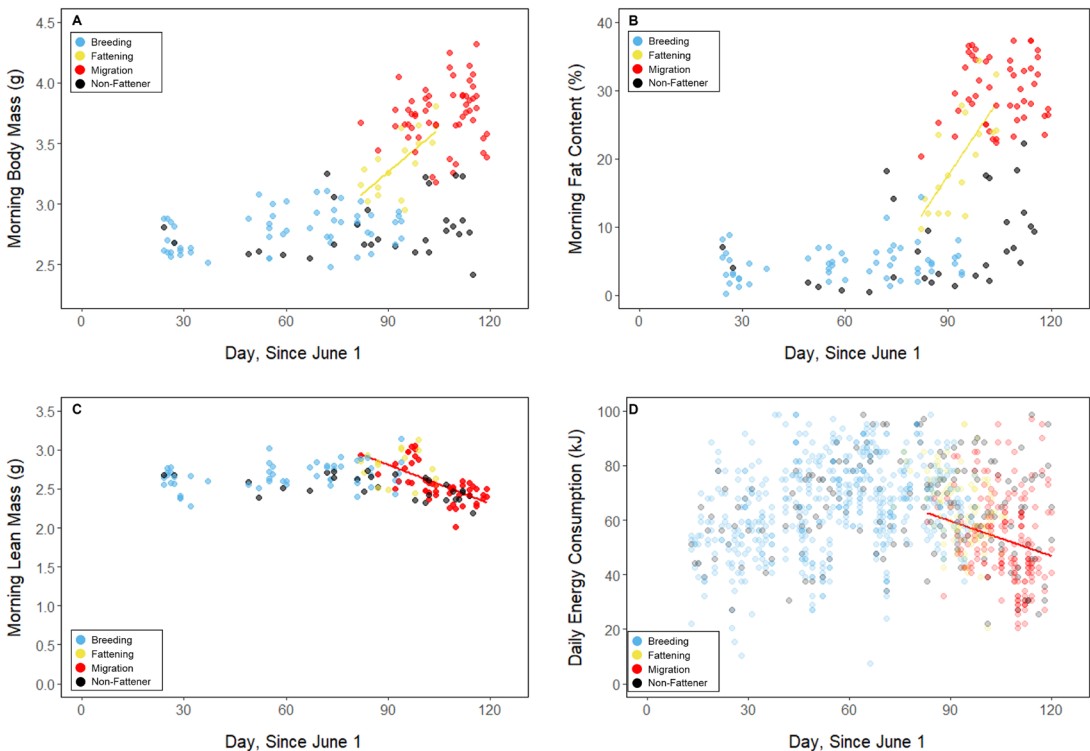

**Appendix 1—figure 5.** Body mass (**A**), fat content (**B**), lean mass (**C**), on the mornings following focal observation nights, and daily energy consumption throughout the study period (**D**), with points and significant trendlines colored by period.

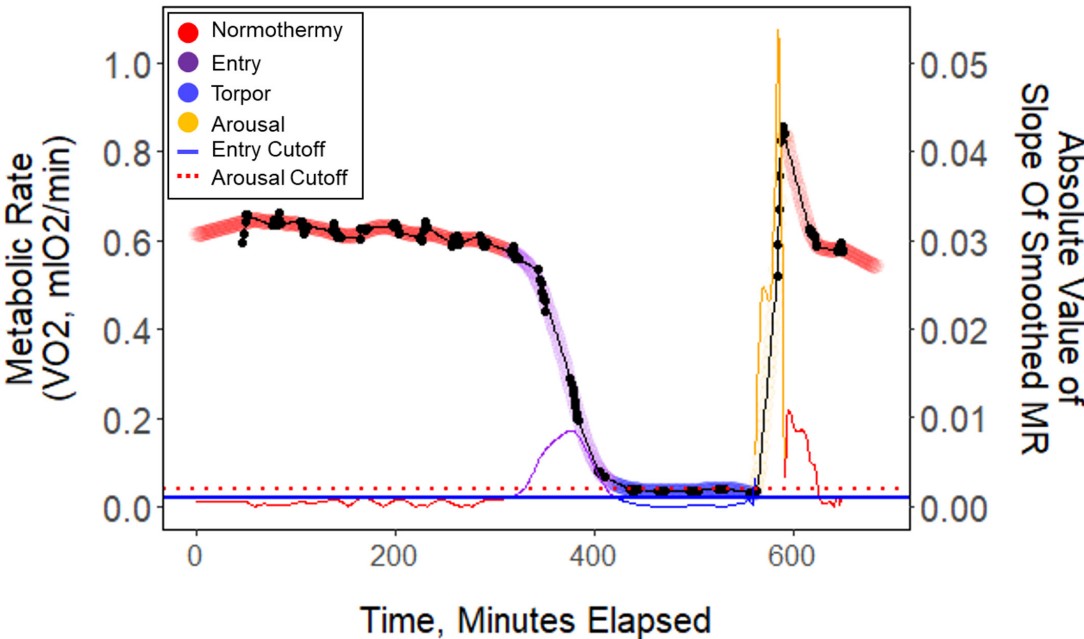

**Appendix 1—figure 6.** Raw $V_{o_2}$ data (black points) plotted every minute throughout one exemplar night of torpor (B13_9.2.19). These data points were smoothed (thick colored line), and the slope of these points (thin colored line) was used to determine metabolic state. Horizontal lines represent entry (blue) and arousal (red) cut-off slopes which were used to distinguish metabolic states of each minute.

