## [Editor Report]

The authors tested the hypothesis that individual hummingbirds employ torpor as an energy-saving mechanism to facilitate migratory fattening, even when starvation is not imminent. This is a difficult hypothesis to test because of the difficulties associated with repeatedly and accurately measuring body composition in individual migratory birds. Using captive experiments, the authors provide some indication that hummingbirds use torpor during energy emergencies in the summer – rare empirical evidence supporting the hypothesis that hummingbirds use torpor to facilitate migratory fitting.

---

## [Decision Letter]

**Decision letter after peer review:**

Thank you for submitting your article "Reversal of the adipostat control of torpor during migration in hummingbirds" for consideration by *eLife*. Your article has been reviewed by 3 peer reviewers, one of whom is a member of our Board of Reviewing Editors, and the evaluation has been overseen by Detlef Weigel as the Senior Editor. The reviewers have opted to remain anonymous.

Your study explored the rules governing the use of torpor and whether they differed between annual cycle stages in the ruby-throated hummingbird. You evaluated changes in the relationships between evening body fat content and torpor occurrence, torpor duration, time of torpor entry, and amount of energy expended before torpor entry during the breeding, fattening and migration periods. Your results suggest that birds shift the rules on when to enter torpor depending on whether they are breeding or during migration. During breeding, ruby-throated hummingbirds go into torpor when body fat stores drop to 5%, whereas during migration they are more likely to enter torpor at high fat levels. You conclude that your findings demonstrate the versatility of torpor as an energy management mechanism throughout the annual cycle.

All the reviews show enthusiasm about the potential of your study to advance our understanding of why, when, and how wild birds use daily torpor as an energetic strategy. The use of respirometry and QMR, and the exemplary job of providing the original data received commendations. You will find this related in the summary evaluation and public review of your work. However, the manuscript will require some major revisions before it can be considered for publication in *eLife*. Particularly, the significance of the results is somewhat overstated because of insufficient recognition of the limitations of the experimental design. The limitations of the study need to be carefully addressed. Please do take note of the following:

Essential revisions:

1. Note that although the controlled conditions were necessary, it presents both a strength and a weakness. The weakness is that natural environmental cues that may affect torpor like variation in temperature, precipitation, and food availability were not in effect. Body mass and composition was convincingly important for torpor initiation and length, but it was difficult to tell what factors caused variation in body mass and composition because all the birds had ad libitum food. Was it just individual variation? Were some birds more stressed than others by captivity? These points need to be carried along in your interpretation and discussion of the results.

2. Following on point one, inter-individual differences should be incorporated not just in models, but the resulting output should be interpreted and discussed. The data shown in figure six and seven of the supplementary information reveals very strong individual patterns. Some individuals may be more likely to go into torpor compared to others regardless of the identified conditions. The three non-fattening birds are a typical example of individuals that do not conform to the norm.

3. Ambient temperature variation is largely ignored here. I found one place where it was indicated that it was measured (line 311), but no place where it was reported. This appears to have been an oversight because the energetics of torpor change dramatically with ambient temperature. It needs to be addressed here from a proximate perspective (what temperature regimes were experienced by the experimental birds and what effects would that be expected to have?) and an ultimate perspective (the shift in 'rules' may be an evolutionary evolved one because of the tendency for lower nighttime temperatures during the early fall), leading to the higher – 65% – frequency of torpor – and in fact, the logistic regression for torpor use should perhaps be reconsidered as far as the way it may have accounted for calendar date.

4. There are conspicuously missing analyses, interpretations, and discussions around how the absolute duration of the night may associate with torpor initiation, duration, and critical thresholds of energy stores. Day length (and therefore, night length) differs between the breeding, fattening, and migration periods and this may influence energy storage and depletion decisions. Could longer nights cause birds to enter torpor earlier and stay longer? Does this affect how much body reserve is accumulated pre-torpor? Differences in night temperature between periods of the annual cycle may also play a role.

5. A general restructuring of the manuscript may help improve the flow. Perhaps separating the results and the discussions will also help to organise things a bit better. This will allow us to see the contrasts in the results quicker because they will be less intertwined with the discussions. I will strongly recommend including a table to show the similarities and the contrasts in the results between the breeding and the migration periods side by side. As it is, the comparisons are difficult to follow.

6. I would like to see the overnight loss of body mass in normothermic birds. Did you measure the overnight fat mass loss during the different periods of the annual cycle (breeding, fattening and migration) for birds that are normothermic? This may be important for determining why birds enter torpor at different fat levels in different periods. The absolute difference in fat mass between periods may not be very important without knowing how much fat is needed to survive the night without torpor.

7. Line 228: The conclusion needs to be toned down. It is not clear that this study has repeatedly and accurately defined a consistent rule governing torpor use in hummingbirds. A detailed examination of the data (supplementary figures 6 and 7) suggests that the occurrence of torpor and the pattern of body mass change over time is specific to individuals. I will strongly recommend that individual patterns be explored. Some individuals may go into torpor regardless of whether the body fat threshold is reached or not. The three fattening birds are a typical example of inter-individual variation.

---

## [Author Response]

Essential revisions:1. Note that although the controlled conditions were necessary, it presents both a strength and a weakness. The weakness is that natural environmental cues that may affect torpor like variation in temperature, precipitation, and food availability were not in effect. Body mass and composition was convincingly important for torpor initiation and length, but it was difficult to tell what factors caused variation in body mass and composition because all the birds had ad libitum food. Was it just individual variation? Were some birds more stressed than others by captivity? These points need to be carried along in your interpretation and discussion of the results.

We agree that variation in temperature, precipitation and food availability would likely affect free-living hummingbirds’ torpor use decisions, and that the fact that we controlled these factors did not allow us to explicitly and directly evaluate their effects on torpor use. However, these environmental factors would most importantly affect the amount of fat the birds start each night with and the rate that they expend those stores. Though we controlled for these factors, individual differences in daily activity and food consumption generated sufficient variation in evening body composition to evaluate how evening energy content was most directly related to torpor use. By controlling potential environmental factors, we greatly simplified the analyses and interpretations, and avoided potentially confounding environmental factors, to confidently conclude that fat mass is the most proximate factor for torpor initiation in the breeding season, and that night length was the primary driver of torpor duration in the migration season. We have added a sentence (134-137) specifying that variation in evening fat content was generated by daily food consumption and activity, and we have added a few sentences (282-292) to discuss the potential effects of factors that we did not allow to vary, including temperature and time fasted. We also now note (278-282) that we did not observe any clear differences in stress levels of the birds.

2. Following on point one, inter-individual differences should be incorporated not just in models, but the resulting output should be interpreted and discussed. The data shown in figure six and seven of the supplementary information reveals very strong individual patterns. Some individuals may be more likely to go into torpor compared to others regardless of the identified conditions. The three non-fattening birds are a typical example of individuals that do not conform to the norm.

We agree that inter-individual differences in torpor use and fattening are interesting, and we have now explicitly discuss the major differences between fattener and non-fattener birds (331-350), and the reliability of instantaneous fat content in relation to the average energy-emergency threshold as a predictor for torpor entry (269-282). Regarding fattening patterns, one important limitation is that it is impossible to confidently assign migratory strategies because the birds were not allowed to actually migrate. For example, it is possible that in a natural context, the birds would have fattened to different extents over different amounts of time. That said, we have now clearly discussed the major difference between non-fatteners and fatteners, and we also describe the important relationship between the amount and duration of fattening with average torpor use and daily food consumption, whereby fattener birds with lower average torpor durations fattened less. We also note that variation in wing morphology (due to wing wear) and pectoral mass (due to activity level) could affect power output and limit the amount of fat mass each bird was able to deposit. Regarding torpor use patterns, we now highlight the consistency of the energy store threshold prediction, whereby only a small portion (15%) did not follow the rule, and we discuss potential reasons for this (i.e. the threshold rules may shift on consecutive nights of torpor use).

3. Ambient temperature variation is largely ignored here. I found one place where it was indicated that it was measured (line 311), but no place where it was reported. This appears to have been an oversight because the energetics of torpor change dramatically with ambient temperature. It needs to be addressed here from a proximate perspective (what temperature regimes were experienced by the experimental birds and what effects would that be expected to have?) and an ultimate perspective (the shift in 'rules' may be an evolutionary evolved one because of the tendency for lower nighttime temperatures during the early fall), leading to the higher – 65% – frequency of torpor – and in fact, the logistic regression for torpor use should perhaps be reconsidered as far as the way it may have accounted for calendar date.

We agree that ambient temperature could be an important variable that affects the decision-making paradigm that governs free-living hummingbirds use of torpor. We did specify that all nighttime temperatures were 20°C, and we now also state that daytime temperatures were also 20°C. We also now explicitly discuss the potential role of ambient temperature (282-293).

4. There are conspicuously missing analyses, interpretations, and discussions around how the absolute duration of the night may associate with torpor initiation, duration, and critical thresholds of energy stores. Day length (and therefore, night length) differs between the breeding, fattening, and migration periods and this may influence energy storage and depletion decisions. Could longer nights cause birds to enter torpor earlier and stay longer? Does this affect how much body reserve is accumulated pre-torpor? Differences in night temperature between periods of the annual cycle may also play a role.

We thank the reviewer for encouraging us to examine the effect of night length more closely. We have adjusted our statistical models that predict torpor use variables (i.e. torpor duration, entry time, critical fat level, energy expenditure before torpor and overnight) to incorporate the interaction between period and night length, in addition to the interaction between period and evening fat mass. Most notably, we think our story is strengthened by results that night length did not influence torpor duration in the breeding period (when evening fat content was the best predictor), and that in the migration period night length, and not evening fat content, did affect torpor duration. This is an important piece of information that further describes the clear switch in torpor use strategies, and it highlights the point that in the breeding period birds simply entered torpor at whatever time they reached critical fat stores, whereas in the migration period, birds entered torpor at consistent times, irrespective of evening fat or a threshold, and the time that they remained in torpor was driven by the length of the night. This allows us to conclude that there is a clear shift in the torpor use rules, from an energy emergency threshold, to a more routine use of torpor. We have adjusted the text throughout the manuscript accordingly.

Regarding night temperatures between periods, there were no differences – as described above, all daytime and nighttime temperatures were 20°C (96, 377, 431).

5. A general restructuring of the manuscript may help improve the flow. Perhaps separating the results and the discussions will also help to organise things a bit better. This will allow us to see the contrasts in the results quicker because they will be less intertwined with the discussions. I will strongly recommend including a table to show the similarities and the contrasts in the results between the breeding and the migration periods side by side. As it is, the comparisons are difficult to follow.

We agree that the paper flows more clearly with separate results and Discussion sections and have reformatted them accordingly.

We also agree that the contrasts between the breeding and the migration seasons are hard to follow while reading the text, and that showing them in a table side by side makes the important takeaways much more clear. Accordingly, we have added a table showing morning body mass, fat content at torpor entry, and the primary drivers (either evening fat content or night length) of the various torpor use metrics with the direction of that relationship.

6. I would like to see the overnight loss of body mass in normothermic birds. Did you measure the overnight fat mass loss during the different periods of the annual cycle (breeding, fattening and migration) for birds that are normothermic? This may be important for determining why birds enter torpor at different fat levels in different periods. The absolute difference in fat mass between periods may not be very important without knowing how much fat is needed to survive the night without torpor.

We did measure overnight mass loss on normothermic nights, and we display this in supplementary figure 4 (now Appendix 1- Figure 4), as open circles. We had included the overall mean ± sem, and have now updated this to be period-specific (216-218). These values are not significantly different (p>0.745) when taking night length into account. We have also calculated and included the threshold fat levels in terms of absolute fat masses in addition to relative fat content (165, 244).

7. Line 228: The conclusion needs to be toned down. It is not clear that this study has repeatedly and accurately defined a consistent rule governing torpor use in hummingbirds. A detailed examination of the data (supplementary figures 6 and 7) suggests that the occurrence of torpor and the pattern of body mass change over time is specific to individuals. I will strongly recommend that individual patterns be explored. Some individuals may go into torpor regardless of whether the body fat threshold is reached or not. The three fattening birds are a typical example of inter-individual variation.

We agree the phrasing of this was not clear with respect to the use of “repeatedly”. We originally meant that we are the first study to make repeated measurements of fat content at torpor entry in the same individuals over time, as previous studies had either used terminal methods and only had one point per bird, or measured only body mass in the same individuals. We have adjusted the phrasing to be more clear (351-352). We have addressed individual variation above.